# Structural Analysis of the Novel Variants of SARS-CoV-2 and Forecasting in North America

**DOI:** 10.3390/v13050930

**Published:** 2021-05-17

**Authors:** Elena Quinonez, Majid Vahed, Abdolrazagh Hashemi Shahraki, Mehdi Mirsaeidi

**Affiliations:** 1Department of Medicine, University of Miami Miller School of Medicine, Miami, FL 33146, USA; exq30@miami.edu; 2Division of Pulmonary, Allergy, Critical Care and Sleep Medicine, University of Miami Miller School of Medicine, Miami, FL 33146, USA; pc.vahed@gmail.com (M.V.); axh1558@med.miami.edu (A.H.S.)

**Keywords:** SARS-CoV-2, S-RBD, mutation, vaccine, COVID-19

## Abstract

Background: little is known about the forecasting of new variants of SARS-COV-2 in North America and the interaction of variants with vaccine-derived neutralizing antibodies. Methods: the affinity scores of the spike receptor-binding domain (S-RBD) of B.1.1.7, B. 1.351, B.1.617, and P.1 variants in interaction with the neutralizing antibody (CV30 isolated from a patient), and human angiotensin-converting enzyme 2 (hACE2) receptor were predicted using the template-based computational modeling. From the Nextstrain global database, we identified prevalent mutations of S-RBD of SARS-CoV-2 from December 2019 to April 2021. Pre- and post-vaccination time series forecasting models were developed based on the prediction of neutralizing antibody affinity scores for S-RBD of the variants. Results: the proportion of the B.1.1.7 variant in North America is growing rapidly, but the rate will reduce due to high affinity (~90%) to the neutralizing antibody once herd immunity is reached. Currently, the rates of isolation of B. 1.351, B.1.617, and P.1 variants are slowly increasing in North America. Herd immunity is able to relatively control these variants due to their low affinity (~70%) to the neutralizing antibody. The S-RBD of B.1.617 has a 110% increased affinity score to the human angiotensin-converting enzyme 2 (hACE2) in comparison to the wild-type structure, making it highly infectious. Conclusion: The newly emerged B.1.351, B.1.617, and P.1 variants escape from vaccine-induced neutralizing immunity and continue circulating in North America in post- herd immunity era. Our study strongly suggests that a third dose of vaccine is urgently needed to cover novel variants with affinity scores (equal or less than 70%) to eliminate developing viral mutations and reduce transmission rates.

## 1. Introduction

The family Coronaviridae contains important zoonotic pathogens, such as SARS-CoV1, MERS-COV, and SARS-CoV-2, which can cause mild to severe respiratory infections in humans [1]. These virions are among the largest RNA viruses and are characterized by their roughly spherical shape with the large spike receptor-binding domain (S-RBD), including glycoproteins that extend 16–21 nm from the viral envelope. The S-RBD of SARS-CoV-2, which interacts with the human angiotensin-converting enzyme 2 (hACE2) receptor to mediate entry into cells [2,3], has been the main target for vaccine development [4]. Routine surveillance of the genomic profile of the SARS-CoV-2 is crucial to discovering relationships between developing viral mutations and transmission rates, vaccine efficacy, and epidemiological tracing.

Several newly emerging variants are circulating the world, raising concerns about their impact on infectivity and mortality, as well as the effectiveness of currently developed vaccines. In South Africa (S.A), a new variant of SARS-CoV-2 (known as 501Y.V2 or B.1.351) has emerged, which carries three mutations at important locations in the S-RBD (K417N, E484K, and N501Y) [5]. The B.1.351 variant represents approximately 36% of the African subsample [6]. Evolution of B.1.351 resulted to more than 16-fold increase of COVID-19 incidence in Zambia [7]. The possibility of a similar experience in the U.S. and other countries is a real threat [8]. A second new variant, known as 501Y.V3 or P.1, carries three mutations in the S-RBD domain (K417T, E484K, and N501Y) and was discovered for the first time in Brazil [9]. It was then detected in other countries including the U.S. [10,11]. A third variant, called 202012/01 (also known as 501Y.V1 or B.1.1.7) was first identified in the United Kingdom (U.K.) [12], and soon after became the primary emerging variant in many countries [13,14,15,16,17,18]. Another new variant which has been designated a Variant Under Investigation (VUI) by Public Health England (PHE); also called B.1.617 variant that was first detected in India on October 5, 2020 [19,20].

Epidemiological analysis has shown that B.1.1.7 is more highly transmissible (50 to 70% more) than other SARS-COV-2 variants [12,21]. The N501Y mutation is shared between all three variants (B.1.1.7, B.1.351, and P.1) but the first lineage (B.1.1.7 variant) has several additional ‘signature’ mutations in the SARS-CoV-2 S proteins [22,23,24]. No alteration has been detected at position 501 of the B.1.617 variant; however, a number of other mutations including E484Q, L452R, and P681R [20] have been identified in this novel variant (Appendix A).

The emergence of mutations in position 501 and 484 is a major concern because it involves one of the six key amino acid residues determining a tight interaction of the S-RBD with its receptor (hACE2) [24]. In the U.S., the Centers for Disease Control (CDC) recently predicted that the number of patients with the B.1.1.7 variant will be increasing [18], however deep analysis of trend of the proportional increase of B.1.1.7 and other emerging variants along with the potential underlying mechanism of rapid spreading is mandatory for further decision making. We report here the forecast of B.1.1.7, B.1.351, B.1.617 and P.1 variants’ isolation in the next 12 months with two pre- and post-vaccination induced herd immunity scenarios in North America. Our models were made based on the binding efficacy of the new mutant variants S-RBD to the neutralizing CV30 antibody and hACE2 receptor.

## 2. Materials and Methods

### 2.1. Time Series Forecasting

SARS-CoV-2 genomic sequence data was gathered from the Nextstrain global database (https://nextstrain.org/ accessed on 2 May 2021), and mutation site numbering and genome structure were done using the Wuhan-Hu-1/2019 genomic sequence as reference (wild-type). The database created by the Global Initiative on Sharing All Influenza Data (GISAID) and the analysis tool Nextstrain have enabled the analysis of thousands of whole-genome sequences of SARS-CoV-2 [25]. Our analysis was performed only for mutations involving the S-RBD domain (see Appendix A for more details). The Nextstrain data bank groups the isolates by regions. Once prevalent mutations were identified, the percentages of mutant genomes were calculated from the total regional subsample of isolates for various periods between December 2019 and April 2021. In addition, separate percentages were calculated from the date of the first appearance of each mutation to the present.

The first forecast model was developed with an assumption that vaccination at the current rate will not affect the proportion of B.1.1.7, B.1.351, B.1.617, and P.1 variants in North America and that there will be equal and sustained rates of infectivity (R 1.1). We then conducted another forecasting model with the assumption of reaching herd immunity with mass vaccination in North America in July 2021. Herd immunity is defined if at least 75% of the eligible population is vaccinated and has developed acceptable levels of neutralizing antibodies in their bodies [26]. Although the efficacy of the COVID-19 vaccine and time duration of protection remains to be determined, we assumed 100% efficacy, with a duration of 12 months of protection.

### 2.2. Construction of the Computational Model

Atom coordinates of wild S-RBD in complex with neutralizing CV30 antibody were extracted from the crystallography structure of the Protein Data Bank (RCSB PDB) with the access code 6XE1 [27]. The neutralizing monoclonal CV30 antibody was isolated from a patient infected with SARS-CoV-2, in complex with the receptor S-RBD [28]. We used the CV30 antibody in our modeling as its binding to the concave hACE2 binding epitope of the S-RBD has previously been well characterized [27]. We obtained hACE2 from electron microscopy of 6CS2 for SARS S Glycoprotein-hACE2 complex [23]. The crystal structure of the N501Y mutant S-RBD of SARS-CoV-2 Spike was obtained from RCSB with the access code 7NEG. The tertiary structures of the S-RBD mutants of B.1.1.7, B.1.617, B.1.351, and P.1 were predicted using the Phyre2 server [29]. HDOCK Server [30] (http://hdock.phys.hust.edu.cn/ accessed on 2 May 2021) was utilized to examine protein-protein interaction based on a hybrid algorithm of template-based modeling into free docking with default parameters. To prepare structures as input files for a docking run, all water molecules and solvent molecules were removed from the proteins. All of the structures were visualized using PYMOL Chimera software version 1 [31]. The calculation procedure was almost the same as that in our previous work [32,33,34] with neutralization. Antibodies were separated from SARS-CoV-2 S-RBD into individual files for docking simulation. The root-mean-square deviation (RMSD) was calculated to evaluate the change in the geometric structure of S-RBD after docking by using backbone atoms. RMSD was used to quantify the similarity between two superimposed atomic coordinates of the reference structure (PDB structure) and the post-structure complex with receptor molecules. The RMSD was calculated to evaluate changes in the geometric structure of the spike protein using spinal atoms.

### 2.3. Analyzed Structures

The number of interface residue pair-wise contacts for B.1.1.7, B. 1.351, B.1.617, P.1 and wild-type complexes with antibody/hACE2 were found with the COCOMAPS web tool with a cut-off distance value of 8 Å^2^ [35]. Two residues of docking are considered in contact if a pair of (any) atoms belonging to two residues are closer than a defined cut-off distance. The advantages of using contact map representations for protein–protein interfaces have also been shown in a previous study [36].

## 3. Results

SARS-CoV-2 variant B.1.351 was tracked for the first time in South Africa in late August 2020, a few days before the emergence of P.1 in Brazil (Figure 1a). Our analysis also shows that B.1.1.7 was detected in the U.K. in early December 2020. The first mutations at positions 501 and 484 were isolated from Côte d’Ivoire; West Africa. Tracking the variants in the U.S. (Figure 1b) also revealed that B.1.1.7 emerged at the end of 2020 (late December), while B.1.351 and P.1 both emerged around the same time, at the end of January 2021. The B.1.617 variant emerged in India in early October 2020 (Figure 1a). This variant was first detected in North America in late February 2021 (Figure 1b).

The forecasting analysis of novel variants in North America subsamples showed that the frequencies of B.1.1.7 (40%), B.1.617 (2%), and P.1 (4%) are still increasing from the times of their emergence until the middle of April (2021); however, the frequency of B.1.351 (2%) shows a noticeable decrease on the same timeline (Appendix A).

Forecast analysis data for models 1 and 2 are shown in Figure 2 for all four novel variants (B.1.1.7, B.1.351, B.1.617 and P.1). Per model 1 (pre-vaccination), B.1.1.7 would become the dominant variant in North America while the rate of others would increase. Though their rates would increase, they would not become the dominant variant (Figure 2a). However, in model 2 (post-vaccination), the rate of B.1.1.7 will sharply decrease (close to zero), but the frequencies of B.1.351, B.1.617, and P.1 variants will gradually increase to 5% after herd immunity is achieved (Figure 2b).

Two residues, N501Y and 484K, from the B.1.351 mutant S-RBD interact with residues from the light chain of the CV30 antibody. The aligned structure of wild-type S-RBD showed that the ranking binding mode (Figure 3), corresponding to the best-scored, did not induce any significant conformational changes in the S-RBD from the CV30 antibody-bound complex. The aligned RBD had an RMSD of 2.32 Å^2^ based on the docking structure. The S-RBD mutant showed significant conformational changes by RMSD of 324.79 Å^2^ (Table 1). Three residues of B.1.1.7 (484E, 417K, and 501Y) have shown hydrogen bonding with hACE2, but lower than wild S-RBD with two residues (484E and 417K) (Figure 4). Residue 417K from the wild S-RBD interacts with 52Y CV30 antibody in the domain of both chains (light and heavy). However, residues 501Y and 484K mutant from B.1.351 mutant interact with light chain S52 and D70 respectively (Figure 5). We also presented docking structures of the interaction of N501Y, E484K, and K417T mutations (P.1 variant) (Figure 6) and E484K only (Appendix A) in complex with a potent neutralizing CV30 antibody/hACE2 (Appendix A). We also presented docking structures of the interaction of E484Q, L452R, and P681R mutations (B.1.617 variant) (Figure 7) in complex with a potent neutralizing CV30 antibody/hACE2. Superposition of wild/mutant SARS-CoV-2 S-RBD protein structures is also shown in Appendix A.

The docking structure in the interaction between SARS-CoV-2 S-RBD protein and Fab of CV30 antibody shows a rough binding affinity for the wild-type S with score affinities of CV30 antibody and S-RBD of the B.1.1.7, B.1.351, P.1, and B.1.617 found to be 90.93%, 70.05%, 72.85%, and 72.53% respectively in comparison to wild-type as shown in Table 1.

The prediction model for overlap between wild and mutant S-RBD proteins with human hACE2 demonstrated a difference between variants as shown in Table 1 and Figure 3. Noting a 100% affinity score for wild-type, B1.1.7, B.1.351, P.1, and B.1.617 demonstrated 114%, 86%, 108%, and 110% affinities, respectively.

In Appendix A, distance range contact maps are shown for the two interfaces of the complex. COCOMAPS property maps report the contacts colored according to the physicochemical nature of involved residues, while the distance range contact maps report the contacts at increasing distances. In Figure 1, it is shown that the binding affinities of the complex WT-CV30 and B.1.1.7-CV30 were stronger compared to the P.1.351-CV30 and P.1-CV30 complexes.

The wild-type and B.1.1.7 variants were highly attached by CV30, even contacted in buried areas and the complex formation residues. However, other variants except B1, antibody bindings were established in none-fragment antigen binding sites. From the COCOMAPS tables, we see indeed that the wild-type interface area is 2010.8 Å2 and 1850.4 Å2 for the Indian variant, where buried residues were the lowest (approximately 92%). These results confirmed that there is a difference in structural properties at RBD area between the two proteins (Table 2). Meanwhile, the analysis of hACE2 showed the Indian interface with CV30 is more extended and that, in it, there are contacts between buried areas and the complex formation residues. We see indeed that the India interface area is 1986.4 Å2 in maximum rate with non polar buried area upon the complex formation (Å2) (Table 3).

## 4. Discussion

The current study shows that the first mutations at positions 501 and 484 were tracked at the same time (July 2020) from the same place (Côte d’Ivoire; West Africa). This finding indicates that the two positions (501 and 484) in S-RBD of SARS-CoV-2 may have a critical role in virus adaptation and evolution during the pandemic. E484K has been reported from the south, northeast, and north Brazilian regions in late August 2020 in the emergence of the P.1 variant [9]. Development of the E484K mutation (P.1 lineage) in Brazil (Rio de Janeiro) [9] and B.1.351 variant in South Africa in late August [5], indicates that position a.a. 484 may be a top hot spot for infectivity, immune escape, and reinfection. The N501T substitution, which probably facilitated the N501Y substitution at position 501, emerged in early July 2020 at Côte d’Ivoire (West Africa) and was then detected in the U.S. in late August 2020. This indicates that SARS-CoV-2 variants carrying mutations at position 501 might have circulated unnoticed before the rapidly emerging lineages such as B.1.1.7, B.1.351, and P.1 (carrying the N501Y mutation) [5,23].

As of January 13, 2021, approximately 76 cases of B.1.1.7 have been detected in different states of the U.S. [18]. B.1.351 and P.1 have also been detected at the end of January 2021 [37,38]. Our timeline analysis also confirmed that variants carrying N501Y mutation (B.1.1.7, B.1.351, and P.1) have emerged between the end of 2020 and the beginning of 2021 in the U.S. (Figure 1). Analysis of SARS-CoV-2 genotypes from California revealed that a new variant called B.1.429 (CAL.20C) was first observed in July 2020 in 1 out of 1247 samples from Los Angeles County; thereafter, this variant’s prevalence has increased to 35% and 44% in California state and Southern California respectively in January 2021 [39]. A recent study reported the presence of other important SARS-CoV-2 variants, but at low rates (B.1.1.7; 23 cases, B.1.351; 2 cases, P.1; 4 cases and two closely related Cal.20C variants; B.1.429; 143 cases and B.1.427; 19 cases) among 20,453 virus specimens from COVID-19 patients dating from December 2020 through mid-February 2021, in the Houston metropolitan region [40]. The novel variant B.1.617 emerged at the beginning of October 2020 from India and was detected in late February 2021 in the U.S. (Figure 1), which is also consistent with the first report of this variant [19,20].

Our forecast analysis predicted that the N501Y mutant (B.1.1.7 lineage) will potentially be the most dominant variant circulating in North America in the coming months if mass vaccination plan is stopped for any reason. However, due to the highest affinity of the neutralizing antibody to this variant (90.93%) relative to the others (70–73% for B.1.351, B.1.617, and P.1), the frequency of this variant will be significantly reduced (a rate of nearly zero). According to the low affinity of neutralizing antibody to B.1.351, P.1 and B.1.617 variants (30% less relative to wild type), the rate of these variants will increase in the community even when herd immunity achieved (Figure 2b). The confirmed cases infected with the B.1.1.7 variant are rapidly increasing; there are 20,915 confirmed B.1.1.7 variant in the U.S, alone, but the rates of other variants, such as B.1.351 (453 cases) and P.1 (497 cases) are not increasing as rapidly (https://www.cdc.gov/coronavirus/2019-ncov/transmission/variant-cases.html, accessed on 5 February 2021). Despite the current increasing rate of B.1.1.7, our analysis shows that in herd immunity era, the frequency of B.1.1.7 will sharply decrease.

CV30 antibody binds to the RBD and competes binding with hACE2. It has been reported that minimal affinity maturation of CV30 antibody significantly impacted the ability of this mAb to neutralize SARS-CoV-2. The CV30 neutralizes the virus by preventing the binding of hACE2 to RBD by direct steric interactions. CV30 also induces shedding of the S1 subunit, indicating an additional mechanism of neutralization [24]. Other mAbs such as CR3022 have been found to cross-bind SARS-CoV-2, but the neutralization was not reported. It has been reported that this mAb was not able to induce shedding of the S protein [24].

Nelson et al., simulated the molecular dynamics of different mutations in the B.1.351 variant and found that the E484K mutation enhances S-RBD affinity for hACE2 and a combination of all three mutations, E484K, K417N, and N501Y, induces conformational change greater than the change induced with the N501Y mutant alone [41]. However, our analysis revealed that while B.1.351 variant has ~14% lower affinity (85.99%) to the hACE2 receptor relative to the wild-type, the B.1.1.7 variant has a 14% higher affinity (113.93%) to this receptor relative to the wild-type (Table 1). These discrepant results most likely occurred due to different affinities of modeled neutralizing antibodies.

Notably, the allosteric effects of B.1.351 S-RBD mutant with neutralizing CV30 antibody could play an important role in immune evasion. Between the binding of the S-RBD protein of SARS-CoV-2 and hACE2, the wild-type shows that the binding free with a small RMSD. However, there must be an important contribution of RMSD to the entropy upon binding. This means that mutant S-RBDs with high RMSDs must overcome much more entropy penalty than wild S-RBD when binding to hACE2; energy is required for a system to change RMSD.

B.1.351 and B.1.1.7 variants showed a significant reduction (~30% and 9% respectively) in their affinities to the neutralizing antibodies relative to wild type, indicating that N501Y confers escaping ability from the neutralizing antibodies to the new variants. The B.1.351 variant could escape from neutralizing antibodies isolated from COVID-19 donor plasma as well as neutralizing CV30 antibody [42,43]. Our analysis showed the chance of escape from neutralizing antibodies could be high (~30%) for B.1.351 and P.1 variants. Similar to Nelson et al. [41], our analysis also showed that the combination of E484K, K417N, and N501Y results in the highest degree of conformational alterations to S-RBD, when bound to neutralized CV30 antibody, compared to wild-type S-RBD. The variant carrying E484K (B.1.351 variant) can escape neutralization by existing first-wave anti-SARS-CoV-2 antibodies and re-infect COVID-19 convalescent individuals [41]. B.1.617 also has E484Q mutation, which can confer escape from neutralization.

Wang et al., showed that the mutants carrying the N501Y mutation (such as B.1.1.7) are relatively resistant to a few mAbs targeting S-RBD and not more resistant to convalescent plasma or vaccine sera, potentially due to lack of the E484K mutation, whereas variants carrying mutations (such as B.1.351, which carries the E484K mutation) are resistant to multiple mAbs targeting S-RBD and are markedly more resistant to neutralization by convalescent plasma (9.4 fold) and vaccine sera (10.3–12.4 fold) [44]. Another recent study showed that serum samples obtained from vaccinated individuals (15 participants vaccinated by Pfizer-BioNTech vaccine) efficiently neutralized wild-type virus. This study suggested that neutralization of B.1.1.7 and P.1 viruses was roughly equivalent relative to neutralization of wild-type while neutralization of B.1.351 virus was lower but still robust [45].

These data are in line with our analysis showing that emergent variants harboring mutations in S-RBD (E484K, K417N, and N501Y) will pose significant challenges for mAb therapy and threaten the protective efficacy of current vaccines. Our data suggests that a 30% reduction in antibody affinity to mutant variants (B.1.351, B.1.617, and P1) would be translatable to up to a 9-fold increase in viral replication rate. Taken together, these data highlight the prospect of reinfection with mutants carrying N501Y and E484K mutations as antigenically distinct variants and may foreshadow reduced efficacy of current S-RBD -based vaccines in the near future. The combination of E484K, K417N, and N501Y results in the highest degree of conformational alterations in the interaction between S-RBD and human hACE2, compared to either E484K or N501Y alone [41]. Currently, three vaccines are authorized and recommended to prevent COVID-19; Pfizer-BioNTech, Moderna, and Janssen & Janssen (J&J). All three vaccines are designed to encode S-RBD [46,47]. The Moderna COVID-19 vaccine produced neutralizing titers against all key emerging variants tested, including B.1.1.7 and B.1.351, with no significant impact on neutralizing titers against the B.1.1.7 variant but a six-fold reduction in neutralizing titers against the B.1.351 variant relative to prior variants [48]. Despite this reduction, neutralizing titer levels with B.1.351 remain above levels that are expected to be somewhat protective [48]. Another study has also shown that the neutralizing activity against B.1.1.7 was essentially unchanged, but was significantly lower against B.1.351 (12.4 fold, Moderna; 10.3 fold, Pfizer) [44] There is no data available for J&J, as this vaccine was approved later. The Oxford COVID-19 vaccine, which has been approved in the U.K., has also shown a substantially reduced neutralization effect against B.1.351 variant compared to wild-type [49], but more studies are currently underway at the University of Oxford for further evaluation of new mutations on Oxford COVID-19 vaccine efficacy.

Our study alarms health authorities about the B.1.617 variant that is currently mainly circulating in India. This strain has poor affinity to neutralizing antibody and high binding ability to the hACE2 receptor, equipping the virus to infect even vaccinated people and making it highly infectious.

The findings of our study should be considered carefully until further validation in experimental study.

## 5. Conclusions

Based on current massive vaccination and pending herd immunity in the U.S., we predict that the rate of B.1.1.7 lineage is going to sharply decrease (near to zero) in coming months. In contrast, the frequency of other variants, B.1.351, B.1.617, and P.1 will gradually increase, even with the achievement of herd immunity in our nation and potentially other countries due to the decreased affinity of these variants to the neutralizing antibodies. The proposed forecasting data may be applicable for other countries with the same infection rates. The current study supports a third dose of vaccination to overcome the lack of activity against the novel mutant variants in people who have already been vaccinated for wild type SARS-CoV-2. We also propose annual vaccination for novel variants that will develop in unvaccinated countries and spread around the world.

## Figures and Tables

**Figure 1 viruses-13-00930-f001:**
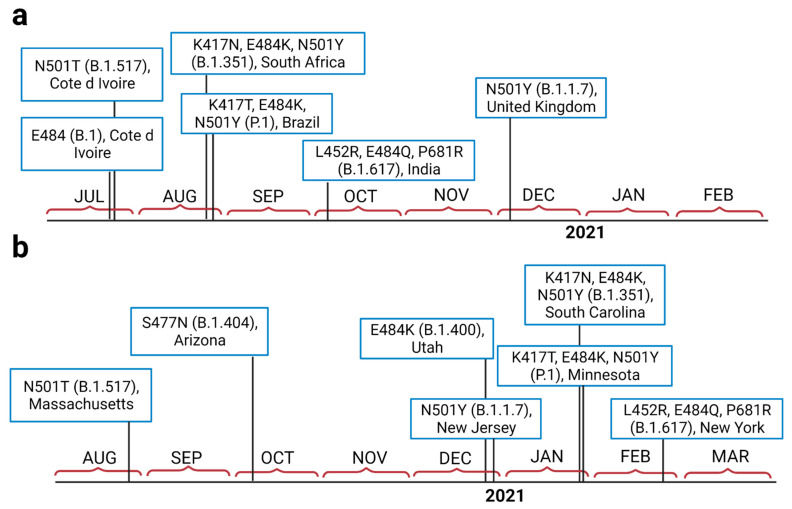
The isolation timeline of currently important emerged variants of SARS-CoV-2in the world (**a**), and in the U.S. (**b**).

**Figure 2 viruses-13-00930-f002:**
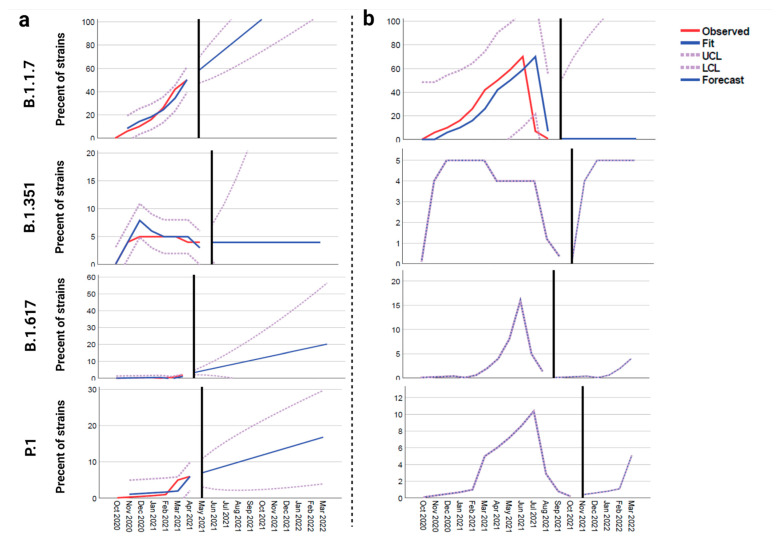
The forecast models of the proportion of different novel SARS-CoV-2 variants in North America. (**a**) Model 1 pre-vaccination, (**b**) model 2 post-vaccination and assumption of occurrence of herd immunity (modeled based on the affinity of neutralizing antibody to the S-RBD protein of B.1.1.7, B.1.617, B.1.351, and P.1 variants).

**Figure 3 viruses-13-00930-f003:**
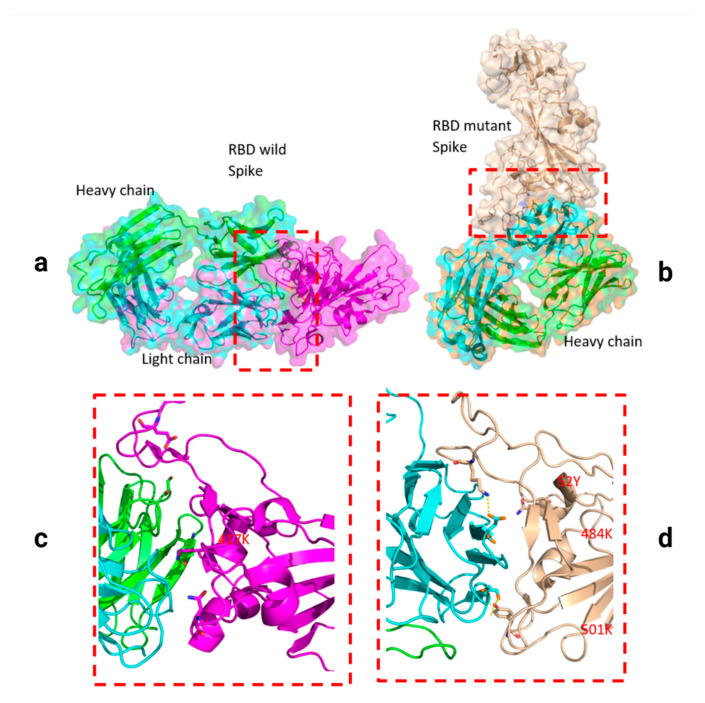
Docking structures of the interaction of SARS-CoV-2 S-RBD protein in complex with a potent neutralizing CV30 antibody. The structure of interactions of the wild S-RBD protein with neutralizing CV30 antibody (**a**,**c**). Docking structure of interactions of mutant S-RBD protein with neutralizing CV30 antibody (**b**,**d**).

**Figure 4 viruses-13-00930-f004:**
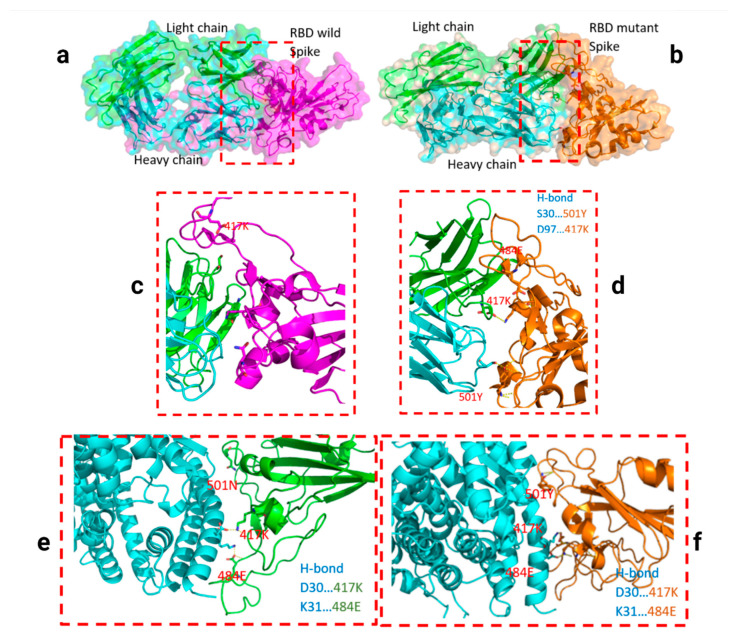
Docking structures of the interaction of SARS-CoV-2 S-RBD protein in complex with a potent neutralizing CV30 antibody. The structure of interactions of wild S-RBD protein with neutralizing CV30 antibody (**a**,**c**). Docking structure of interactions of mutant B.1.1.7 S-RBD protein with neutralizing CV30 antibody (**b**,**d**). The structure of interactions of wild S-RBD protein with human hACE2 (**e**). The structure of interactions of mutant B.1.1.7 S-RBD (N501Y) protein with human hACE2 (**f**). The dashed line indicates the hydrogen bond.

**Figure 5 viruses-13-00930-f005:**
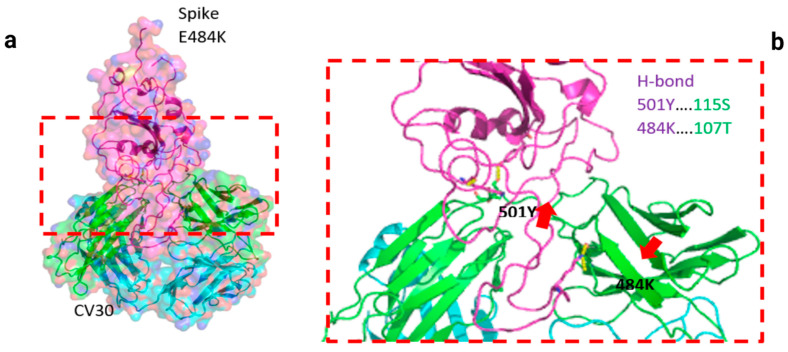
Docking structures of the interaction of B.1.351 mutant SARS-CoV-2 S-RBD protein in complex with a potent neutralizing CV30 antibody (**a**) and hACE2 (**b**). The dashed line indicates the hydrogen bond.

**Figure 6 viruses-13-00930-f006:**
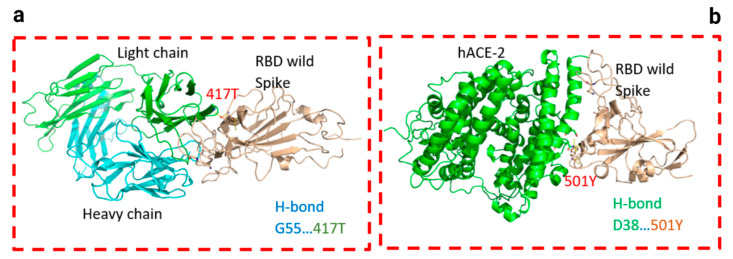
Docking structures of the interaction of P.1 mutant SARS-CoV-2 S-RBD protein in complex with a potent neutralizing CV30 antibody (**a**) and hACE2 (**b**). The dashed line indicates the hydrogen bond.

**Figure 7 viruses-13-00930-f007:**
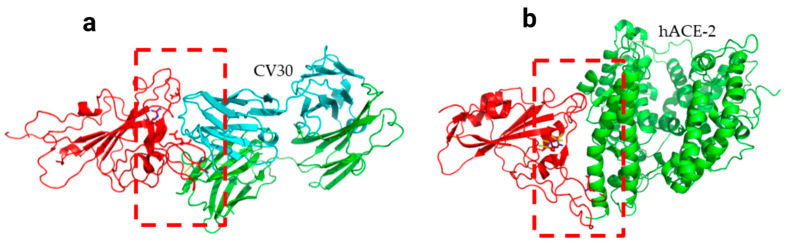
Docking structures of the interaction of B.1.617 mutant SARS-CoV-2 S-RBD protein in complex with a potent neutralizing CV30 antibody (**a**) and hACE2 (**b**). The dashed line indicates the hydrogen bond.

**Table 1 viruses-13-00930-t001:** Docking score of binding of S-RBD region of different novel variants with neutralizing CV30 antibody and hACE2.

	CV30/Spike						hACE2/Spike				
Rank	* WT	B.1.1.7	B.1.351	P.1	E484K	B.1.617	* WT	B.1.1.7	B.1.351	E484K	B.1.617
Docking Score	−371.61	−337.93	−260.34	−270.72	−263.21	−269.54	−310.19	−353.32	−266.76	−323.78	−341.31
Ligand RMSD (Å^2^)	2.32	142.44	324.79	385.41	74.78	384.23	3.27	0.79	93.91	0.56	0.36
Score Affinity %	100	90.93	70.05	72.85	70.82	72.53	100	113.9	85.99	104.38	110.03

* Wild-type (WT) data was considered with a 100% index, and data mutations were measured according to this criterion.

**Table 2 viruses-13-00930-t002:** The interaction features for the area of interfaces antibody (CV30)-spike complexes.

MOL1/MOL2	WT/CV30	B.1.1.7 /CV30	B.1.351 /CV30	484/ CV30	P.1/ CV30	B.1.617/ CV30
Buried area upon the complex formation (Å2)	2010.8	2010.9	1952.8	1977.8	2061.7	1850.4
Buried area upon the complex formation (%)	6.51	6.47	5.69	8.19	6.68	6.01
Interface area (Å2)	1005.4	1005.45	976.4	988.9	1030.85	925.2
Interface area MOL1 (%)	8.69	8.69	6.52	7.74	8.91	8.09
Interface area MOL2 (%)	5.2	5.16	5.05	8.69	5.34	4.79
POLAR Buried area upon the complex formation (Å2)	1084	1084.1	956.2	1149.3	1042.4	930.5
POLAR Interface (%)	53.91	53.91	48.97	58.11	50.56	50.29
POLAR Interface area (Å2)	542	542.05	478.1	574.65	521.2	465.25
NON POLAR Buried area upon the complex formation (Å2)	927	927	996.6	828.5	1019.1	919.9
NON POLAR Interface (%)	46.1	46.1	51.03	41.89	49.43	49.71
NON POLAR Interface area (Å2)	463.5	463.5	498.3	414.25	509.55	459.95
Residues at the interface_TOT (*n*)	256	249	228	66	247	232
Residues at the interface MOL1	32	32	30	33	28	27
Residues at the interface MOL2	224	217	198	33	219	205

Abbreviation: buried area upon the complex formation (Å2): the buried ASA in the complex (free ASA minus complex ASA). Buried area upon the complex formation (%): percentage of buried ASA in the complex (buried area divided by free ASA, percentage). Interface Area (Å2): the area involved in the interaction (half of the buried area upon the complex formation). Interface Area MOL1 (%): the percentage of ASA used by Molecule1 in the complex formation (Molecule1 interface area divided by Molecule1 free ASA, percentage). Interface Area MOL2 (%): the percentage of ASA used by Molecule2 in the complex formation (Molecule2 interface area divided by Molecule2 free ASA, percentage). POLAR buried area upon the complex formation (Å2): the POLAR buried ASAin the complex (free POLAR ASA minus complex POLAR ASA). POLAR interface area (%): percentage of POLAR contribution to the interface (POLAR buried ASA divided by total buried ASA, percentage). POLAR Interface Area (Å2): The POLAR area involved in the interaction (half of the POLAR buried area upon the complex formation). NON POLAR buried area upon the complex formation (Å2): the NON POLAR buried ASAin the complex (free non polar ASA minus complex NON POLAR ASA). NON POLAR interface area (%): Percentage of NON POLAR contribution to the interface (NON POLAR buried ASA divided by total buried ASA, percentage). NON POLAR Interface Area (Å2): the NON POLAR area involved in the interaction (half of the NON POLAR buried area upon the complex formation). Residues at the interface_TOT (%): percentage of residues involved in the complex formation (number of buried residues divided by the total number of residues, percentage). Residues at the interface_MOL1 (%): percentage of residues involved in the complex formation (number of Molecule1 buried residues divided by the total number of Molecule1 residues, percentage). Residues at the interface_MOL2 (%): percentage of residues involved in the complex formation (number of Molecule2 buried residues divided by the total number of Molecule2 residues, percentage).

**Table 3 viruses-13-00930-t003:** The interaction features for the area of interfaces hACE2-spike complexes.

Mol1/Mol2	WT/ hACE2	B.1.1.7 /hACE2	B.1.351 /hACE2	484 /hACE2	P.1/ hACE2	B.1.617 /hACE2
Buried area upon the complex formation (Å2)	1902.5	1955.3	2568.1	1944.6	1894.6	1986.4
Buried area upon the complex formation (%)	5.46	5.63	6.48	5.6	5.45	5.77
Interface area (Å2)	951.25	977.65	1284.05	972.3	947.3	993.2
Interface area MOL1 (%)	9.54	9.78	8.57	9.75	9.52	9.93
Interface area MOL2 (%)	3.82	3.95	5.21	3.93	3.82	4.06
POLAR Buried area upon the complex formation (Å2)	1122.7	1103.8	1572.3	1109.7	1120	1111
POLAR Interface (%)	59.01	56.45	61.22	57.07	59.12	55.93
POLAR Interface area (Å2)	561.35	551.9	786.15	554.85	560	555.5
NON POLAR Buried area upon the complex formation (Å2)	779.9	851.6	995.8	835	774.6	875.5
NON POLAR Interface (%)	40.99	43.55	38.78	42.94	40.88	44.07
NON POLAR Interface area (Å2)	389.95	425.8	497.9	417.5	387.3	437.75
Residues at the interface TOT (*n*)	56	53	78	54	53	58
Residues at the interface MOL1	27	27	35	26	26	29
Residues at the interface MOL2	29	26	43	43	27	29

## Data Availability

All data generated and analyzed during this study are included in this article and its Appendix A. We also extracted SARS-CoV-2 data from Nextstrain database. Nextstrain is an all-source code is freely available under the terms of the GNU Affero General Public License. Nextstrain is supported by NIH for research on pathogens including SARS-CoV-2. We have accessed to de-identifiable data.

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
