# Peer review of "Structural Analysis of the Novel Variants of SARS-CoV-2 and Forecasting in North America"

_viruses, 2021, doi:10.3390/v13050930_

Round 1

Reviewer 1 Report

Quinonez et al predicted the affinity of Spike protein of the new variants of SARS-COV-2 to its neutralizing antibodies and receptor using Nextstrain global database. The authors also reported the forecast of B.1.1.7 variant in the next 12 months with two vaccination efficacy scenarios in the U.S..  This manuscript addressed important questions in the field and could provide potential helpful information.

I have the following concerns and suggestions:

  1. Please add a table to summarize the main mutations, and other information in different SARS-CoV-2 variants. 1.351 (K417N, E484K, and N501Y), P1(K417T, E484K, and N501Y), B.1.1.7(N501Y, A570D, D614G, P681H, T716I, S982A, and D1118H, deletions at 69-70 and 144).
  2. Line 215-217, “Both B.1.351 and B.1.1.7 variants showed a significant reduction (~30%) of their affinity to the neutralizing antibodies relative to wild type,” did not match with Table 1, where B1.1.7 showed only 9.1% of affinity reduction.
  3. If possible, it is interesting to know the affinity score for CV3 and RBD with N501Y only mutation (in table 1).
  4. Line 241-242, the authors highlighted N501Y as a distinct variant to affect the current RBD-based vaccine efficacy, however, based on Table 1, it seems that E484K is the main one that affects the binding affinity to antibody elicited by WT virus. B1.135, P1 and E484K result in 30% of affinity reduction, while B1.1.7 (having N501Y, but not E484K) leads to less than 10% of affinity reduction.
  5. In figure 2, model2, please provide the rationale to make the assumption that vaccine reduced the rate of B1.1.1.7 by 70%. If the prediction of vaccine is based on the score affinity as shown in Table 1, B1.1.7 is 90.93%, which means that vaccine reduced rate should be about 90%.
  6. Based on current publications, it seems that vaccine-mediated protection was more affected by B1.351, P1, rather than B1.1.7. can the authors provide predictions on variants B1.351, P1?

Author Response

Reviewer#1.

I have the following concerns and suggestions:

Quinonez et al predicted the affinity of Spike protein of the new variants of SARS-COV-2 to its neutralizing antibodies and receptor using Nextstrain global database. The authors also reported the forecast of B.1.1.7 variant in the next 12 months with two vaccination efficacy scenarios in the U.S.  This manuscript addressed important questions in the field and could provide potential helpful information.

 Comment: Please add a table to summarize the main mutations, and other information in different SARS-CoV-2 variants. 1.351 (K417N, E484K, and N501Y), P1(K417T, E484K, and N501Y), B.1.1.7(N501Y, A570D, D614G, P681H, T716I, S982A, and D1118H, deletions at 69-70 and 144).

Response: Thanks for comment. We added a table (Supplementary Table 1) summarizing the mutations of different SARS-CoV-2 variants.

Comment: Line 215-217, “Both B.1.351 and B.1.1.7 variants showed a significant reduction (~30%) of their affinity to the neutralizing antibodies relative to wild type,” did not match with Table 1, where B1.1.7 showed only 9.1% of affinity reduction.

Response: We apologize for the mistake. We revised the paper as B.1.1.7 shown only 9.1% of affinity reduction and added forecasts for all other variants. 

Comment: If possible, it is interesting to know the affinity score for CV3 and RBD with N501Y only mutation (in table 1).

Response: SARS-CoV-2 UK variant (B.1.1.7) is only mutation S-RBD (N501Y). In the present study docking affinity score CV30 neutralizing antibodies with RBD with N501Y (90.93%) suggest that the U.K N501Y variant can still be neutralized efficiently by an antibody (Table. 1).

Comment: Line 241-242, the authors highlighted N501Y as a distinct variant to affect the current RBD-based vaccine efficacy, however, based on Table 1, it seems that E484K is the main one that affects the binding affinity to antibody elicited by WT virus. B1.135, P1 and E484K result in 30% of affinity reduction, while B1.1.7 (having N501Y, but not E484K) leads to less than 10% of affinity reduction.

Response: Thanks for comment! We revised the sentence by adding E484K mutant as main mutation affecting the binding affinity to antibody. The revised sentences are: “Our data suggest that a 30% reduction in antibody affinity to mutant strains (B1.135 and P1) would be translatable to up to a 9-fold increase in viral replication rate. Taken together, these data highlight the prospect of reinfection with mutants carrying N501Y and E484K mutations as antigenically distinct variants and may foreshadow reduced efficacy of cur-rent S-RBD -based vaccines in the near future.”

Comment: In figure 2, model2, please provide the rationale to make the assumption that vaccine reduced the rate of B.1.1.1.7 by 70%. If the prediction of vaccine is based on the score affinity as shown in Table 1, B.1.1.7 is 90.93%, which means that vaccine reduced rate should be about 90%.

Response: Thanks again for mentioning it! We revised the paper according to the table 1 data. Moreover, we added forecasting models for B.1.135 and P.1. We also performed the same analysis on B.1.617 as a new variant and added all relevant information to the paper.

Comment: Based on current publications, it seems that vaccine-mediated protection was more affected by B1.351, P1, rather than B1.1.7. can the authors provide predictions on variants B1.351, P1?

Response: Your great comment encouraged us to expand our analysis to other variants. We performed few more analyses to develop the prediction models for emerging variants; B1.351, P1 and B.1.617 and revised the entire paper.

Reviewer 2 Report

General comments

Authors provide an structural basis for the reduced humoral cross-protection empirically observed in some SARS-CoV-2 strains-of-concern. I find the approach is technically correct given that neutralizing antibodies are mostly those binding to the receptor binding domain of the spike protein and B-cell epitope recognition shows often a high structural nature. Thus, epitope switch may be difficult to address by sequence bioinfomatic strategies. This knowledge could also pave the way to interpret vaccine evasion by future emergent lineages. However, I find some further issues may be considered to improve the quality of the study.

Major points

1) I miss a more detailed analysis of the molecular interaction within the framework of the knowledge of antibody:antigen complexes. This could increase the mechanistic support of the conclusions drawn. For instance, some insights may be taken from Kringelum et al, 2013 (PMID: 22784991) or similar papers.

2) I understand the analyses conducted here use structural "static" information. I wonder whether the authors have access to computer facilities to apply some energy minimization and a short molecular dynamics on antibody-antigen complexes for both wild-type and mutant evasive versions. If doable, this may also enhance the reliability of the findings.

3) The humoral protection is based on the production of polyclonal responses consisting on a myriad of different antibodies and recognized epitopes. Authors should emphasize why a simulation using a single monoclonal antibody (while efficient as a biomedical tool) would rationally mirror the humoral evasion actually happening in nature.

Minor points

1) L1-3. I think the title does not represent the work done. This is a two-stage manuscript but I feel most merit is in the structural analysis while lineage tracking is rather preparatory.

2) L72-76. Font size reduction should be corrected.

Author Response

Comments and Suggestions for Authors

General comments

Authors provide an structural basis for the reduced humoral cross-protection empirically observed in some SARS-CoV-2 strains-of-concern. I find the approach is technically correct given that neutralizing antibodies are mostly those binding to the receptor binding domain of the spike protein and B-cell epitope recognition shows often a high structural nature. Thus, epitope switch may be difficult to address by sequence bioinfomatic strategies. This knowledge could also pave the way to interpret vaccine evasion by future emergent lineages. However, I find some further issues may be considered to improve the quality of the study.

Major points

Comment: I miss a more detailed analysis of the molecular interaction within the framework of the knowledge of antibody:antigen complexes. This could increase the mechanistic support of the conclusions drawn. For instance, some insights may be taken from Kringelum et al, 2013 (PMID: 22784991) or similar papers.

Response: Thanks for your great comment! We performed analysis structural using COCOMAPS web tool with a cut-off distance value of 8 Å for all new variants (B.1.1.7, B.1.351. B.1.617 and P.1) and added the information to the paper (Table 2 and 3, Supplementary figure 4).  Our analysis showed that the binding affinities of the complex WT-CV30 and B.1.1.7-CV30 were stronger compared to the B.1.351-CV30 and P.1-CV30 complex. These results confirmed that there is a difference in structural properties at RBD area between the two proteins (Table 2). Meanwhile, the analysis of hACE-2 showed the B.1.351-hACE-2 complex has the no number of Hbond of 417-484-501 compared to other, but the B.1.351 shows more residues and hydrophilic area at the interface area (Table 3).

Comment:  I understand the analyses conducted here use structural "static" information. I wonder whether the authors have access to computer facilities to apply some energy minimization and a short molecular dynamic on antibody-antigen complexes for both wild-type and mutant evasive versions. If doable, this may also enhance the reliability of the findings.

Response: It is great idea to apply some energy minimization and a short molecular dynamic on antibody-antigen complexes for both wild-type and mutant evasive versions, Unfortunately, we are unable to perform it due to our access limitations. We will try to perform dynamic analysis in our future study.

Comment:  The humoral protection is based on the production of polyclonal responses consisting on a myriad of different antibodies and recognized epitopes. Authors should emphasize why a simulation using a single monoclonal antibody (while efficient as a biomedical tool) would rationally mirror the humoral evasion actually happening in nature.

Response: We agree with the reviewer comment regarding the fact that humoral protection is based on the production of different antibodies, however we only focused of CV30 as this antibody studied well in details such as x-ray crystal structure and able to interact with hACE-2  and RBD, has minimal affinity maturation of CV30, and able to induce shedding of the S protein and etc. However, other mAb such as CR3022 found to cross-bind SARS-CoV-2 but the neutralization was not reported and was not able to induce shedding of the S protein.

We added the following sentence to the paper to address the reviewer comment; “Humoral protection is based on the production of polyclonal responses consisting of a myriad of different antibodies and recognized epitopes, however, we used CV30 mAb which is isolated from a patient infected with SARS-CoV-2 for stimulation in this research. This antibody binds to the RBD and competes binding with hACE-2. Moreover, minimal affinity maturation of CV30 antibody significantly impacted the ability of this mAb to neutralize SARS-CoV-2, indicating that CV30 neutralizes the virus by preventing the binding of hACE-2 to RBD by direct steric interactions. CV30 also induces shedding of the S1 subunit, indicating an additional mechanism of neutralization [24]. Other mAb such as CR3022 found to cross-bind SARS-CoV-2 but the neutralization was not reported and is has been reported that this mAb was not able to induce shedding of the S protein [24].

Minor point: I think the title does not represent the work done. This is a two-stage manuscript, but I feel most merit is in the structural analysis while lineage tracking is rather preparatory.

Response: Thanks for your comment. We revised the title to: “Structural analysis of the novel variants of SARS-CoV-2 and forecasting in North America”.

Minor point: Font size reduction should be corrected.

Response: We corrected the font size issue.

Round 2

Reviewer 1 Report

The concerns have been addressed. 

Reviewer 2 Report

I estimate the authors have adequately addressed muy concerns from my previous review.